# A Comprehensive Study on EMI Shielding Performance of Carbon Nanomaterials-Embedded CFRP or GFRP Composites

**DOI:** 10.3390/polym14235224

**Published:** 2022-12-01

**Authors:** Daeik Jang, Bum-Jun Kim, Il-Woo Nam

**Affiliations:** 1Department of Civil and Environmental Engineering, Korea Advanced Institute of Science and Technology (KAIST), 291 Daehak-ro, Yuseong-gu, Daejeon 34141, Republic of Korea; 2School of Spatial Environment System Engineering, Handong Global University, 558 Handong-ro, Buk-gu, Pohang 37554, Republic of Korea

**Keywords:** electromagnetic interference shielding, CFRP and GFRP, carbon nanomaterials, fiber-reinforced plastics, polymer-based composites

## Abstract

The rapid advancement of electrical and telecommunication facilities has resulted in increasing requirements for the development of electromagnetic interference (EMI) shielding composites. Accordingly, an experimental study was conducted to evaluate the EMI shielding performance of carbon nanomaterial (CNM)-embedded carbon-fiber-reinforced polymer (CFRP) or glass-fiber-reinforced polymer (GFRP) composites. Nine combinations of CNMs and carbon or glass fibers were used to fabricate the composites. The synergistic effects of CNMs on the EMI shielding performance were systematically investigated. The results indicated that plate-type CNMs (i.e., graphene and graphite nanoplatelets) have more prominent effects than fiber-type CNMs (carbon nanofibers). The composites fabricated with CFRP afforded higher EMI shielding than the GFRP-based composites. Among the eighteen samples, 3% CNT-GNP in CFRP composites, which included plate-typed CNM, exhibited the best EMI shielding performances, showing 38.6 dB at 0.7 GHz. This study helps understand the shielding performance of CNM-embedded CFRP and GFRP composites in electrical and telecommunication facilities.

## 1. Introduction

With the rapid advancement of electrical devices and telecommunications technology, the importance of electromagnetic interference (EMI) shielding has increased [1,2,3]. Since the undesirable EM wave can cause harmful effects on nearby electronics, it is highly required to develop EMI shielding composites to mitigate the harmful effects [1,2,3]. For these reasons, the developments of EMI shielding composites have been proposed by many researchers [4]. Particularly, the developments of EMI shielding composites using metallic-based materials have been proposed [5]. However, the use of metallic materials can cause corrosion problems and increase the weight of the shielding composites [5,6,7,8,9]. Moreover, metallic material-based EMI shielding composites can generate undesirable interference from EM waves due to the reflection of EM waves at the edge of the shielding composites. Therefore, metallic material-based EMI shielding composites can pose various obstacles in field applications [5,6,7,8,9].

Recently, polymeric composites incorporating carbon-based nanomaterials (CNMs) for EMI shielding have been highlighted [10,11,12]. CNMs with excellent electrical conductivity and outstanding EM wave-shielding properties are regarded as prominent candidates for fabricating EMI shielding composites [10,11,12]. Thus, many researchers have attempted to utilize CNM-embedded polymeric composites as lightweight EMI shielding composites to mitigate the drawbacks of conventional EMI shielding composites [10,11,12]. Yang et al. [10] developed carbon nanotubes (CNTs) and graphene-embedded silicon-based composites for EMI shielding, where the fabricated composites exhibited shielding effectiveness of 32.1 dB in the frequency range of 8.2–12.4 GHz, which is called the X-band. Zhao et al. [11] fabricated hybrid sponge-type EMI shielding composites using reduced graphene oxide (rGO) and CNTs, achieving approximately 35 dB of shielding effectiveness in the X-band. Jang et al. [12] manufactured CNT-embedded polymeric composites with a shielding effectiveness of approximately 22 dB at a frequency of 8 GHz. Qi et al. [13] examined the EMI shielding effectiveness of a polyvinylidene fluoride (PVDF)-based sandwich structure consisting of graphite nanoplatelets (GNPs), Ni, and CNTs [13]. The composite structure afforded approximately 40 dB shielding effectiveness at a frequency of 13 GHz [13]. Wang et al. [14] also fabricated polyimide foam incorporating CNT/graphene oxide (GO), and found that the average EMI shielding effectiveness was 28.2 dB in the frequency range of 8.3–12.3 GHz [14]. Din et al. [15] fabricated GNP paper-inserted GFRP composites, and the composite including GNP paper of 240 μm exhibited EMI shielding effectiveness of 59 dB at 8.5 GHz.

Intensive efforts have been devoted to achieving enhanced EMI shielding performance using various types of CNMs, but few studies have compared the EMI shielding performance when various CNMs are employed to fabricate the shielding composites. In particular, the investigations of EMI shielding performance of carbon/glass-fiber-reinforced polymeric (CFRP and GFRP) composites are rare. These composites have been highlighted due to their light weight, high strength, excellent corrosion resistance, and high fatigue resistance [16,17]; however, few studies have reported the EMI shielding performance of CFRP and/or GFRP composites. In addition, comprehensive studies on the effects of the CNM type on the EMI shielding characteristics of CNM-embedded CFRP and GFRP composites have not been examined to date, to the best of our knowledge. In this regard, this study aims to evaluate the EMI shielding performance of various types of CNM-embedded CFRP and GFRP composites. Nine combinations of CNMs and two types of microfibers (i.e., carbon fibers (CFRP) and glass fibers (GFRP)) are used to fabricate the composites to examine the synergistic effects of various combinations of CNMs on improvements of EMI shielding performances. Based on the EMI shielding performance, the optimized combinations of various types of CNMs and microfibers are analyzed.

## 2. Materials and Methods

### 2.1. Materials

Four types of CNMs—CNTs, carbon nanofibers (CNFs), graphene, and graphite nanoplates (GNPs)—were mixed in CFRP and GFRP composites. The CNTs, CNFs, and graphene were purchased from Daoking Co., Ltd. (Beijing, China), and GNPs were purchased from Timenano Co., Ltd. (Chengdu, China). The epoxy resin (E-4676) and hardener (HC-3008-5) were obtained from Xiangfeng New Composite Co., Ltd. (Kunshan, China) and were mixed in a 3:1 ratio. The dimensions and physical properties of the CNMs are listed in Table 1 [18]. Dimensions, purity, specific surface area, length or width, and electrical conductivity of CNMs shown in Table 1 refer to specifications provided by the manufacturers. Carbon and glass-fiber-woven plain fabrics supplied by Miaohan Construction & Technology Co., Ltd. (Shanghai, China) and Suihua Glass Fiber Co., Ltd. (Jiangxi, China), respectively, were used to fabricate the CNM-embedded CFRP and GFRP composites. The physical properties of the plain carbon/glass-fiber fabrics are listed in Table 2. The grade, thickness, and elongation at the break shown in Table 2 refer to specifications provided by manufacturers. Nine mix proportions of CNMs were embedded into the CFRP and GFRP composites. Pure CFRP and GFRP without incorporating CNMs, CNT-embedded CFRP and GFRP, and hybridized CNMs (combinations of graphene, CNF, GNP with CNT)-embedded CFRP and GFRP composites were prepared. In the hybridized CNM-embedded CFRP and GFRP composites, the ratio of the CNT content to another CNM (graphene, CNF, or GNP) was maintained at 1:1. In addition, the ratio of the gross CNM weight to the total weight, including the epoxy resin, was increased from 1.5% to 3.0%. Details of the mix proportions are presented in Table 3.

### 2.2. Sample Preparation

A three-roll mixer (ZYE-50, Shenzhen Zhong Yi Technology Co., Ltd., Shenzhen, China) was used to improve the dispersion of the CNMs in the epoxy resin. The utilization of three-roll mixer is a simple and fast procedure for dispersing CNMs in a polymer matrix. The three-roll mixer has been used to enhance homogeneous dispersion of nanomaterials in polymer matrix materials, and it has proved an increase in the reliability of the performance and characteristics of the composites [15]. In addition, this method requires a minimal amount of solvent and improves economic efficiency [19,20,21]. However, the three-roll milling process may entail disadvantage such as damages in nano-structures of the CNMs due to repeatedly applied mechanical forces.

A schematic of the synthesis of the CNM-embedded FRP composites is shown in Figure 1.

The required amounts of the CNMs, epoxy resin, and curing agent were weighed. The prepared materials were manually mixed for 2–3 min and subsequently placed in a three-roll mixer. The epoxy resin containing CNMs was squeezed through the two gaps in the three-roll mixer (with a range of 1–150 µm), and the following process was repeated for 12 cycles [15]. The gap distances were carefully adjusted at each cycle, and the gap distance was described as follows. At 1st cycle, the distance of gaps of mixture inlet and outlet were 60 μm and 40 μm, respectively. Then, they were changed to 40 μm and 20 μm at 2nd cycle, 20 μm and 15 μm at 3rd cycle, 15 μm and 10 μm at 4th cycle, 10 μm and 5 μm at 5th cycle, 5 μm and 5 μm at 6th cycle, 10 μm and 5 μm at 7th cycle, 5 μm and 5 μm at 8th cycle, 10 μm and 5 μm at 9th cycle, 5 μm and 5 μm at 10th cycle, 3 μm and 2 μm at 11th cycle, and 2 μm and 1 μm at 12th cycle, respectively.

The dispersed CNM-embedded epoxy resin was coated onto an aluminum plate (320 mm × 320 mm), and the fabric was laid. This process was repeated five times to fabricate a five-layer GFRP and six times to fabricate six-layer CFRP composites. The CNM-embedded CFRP and GFRP were subjected to vacuum infiltration (Shenzhen Airmate Technology Co., Ltd.) to remove trapped air. The fabricated CFRP and GFRP were then cut into rectangular plates with dimensions of 250 mm (length) × 25 mm (width), according to ASTM D 3039.

### 2.3. EMI Shielding Performance Test

A programmable network analyzer (PNA) (Agilent N5239) and 7 mm airline instrument (Agilent 85051) with material measurement software (Agilent 85071E, Agilent Technologies Inc., Santa Clara, CA, USA) were employed to analyze the EMI shielding performance, represented by the *S*-parameters (i.e., *S*_11_, *S*_12_, *S*_21_, *S*_22_), in the frequency range of 300 kHz to 8.5 GHz [12]. The reflection and absorption loss values can be determined according to the observed *S*-parameters [6]. The correlations between the *S*-parameters, reflection, and absorption loss values were previously elaborated by Al-Saleh and Sundararaj [22].

## 3. Results and Discussion

### 3.1. EMI Shielding Effectiveness as a Function of Frequency

The EMI shielding effectiveness of the samples fabricated with different microfibers (i.e., carbon fiber (CFRP) and glass fiber (GFRP)) is shown in Figure 2, where the effects of the microfiber on the EMI shielding capability are demonstrated.

For the CFRP-based samples, the EMI shielding effectiveness was higher than 15 dB, as shown in Figure 2a. This indicates that carbon-fiber-based composites with high electrical conductivity can lead to high EMI shielding effectiveness. In addition, the synergistic effects of the embedded CNMs and the carbon-fiber-based composites can improve the EMI shielding capability. This is elaborated upon in Section 3.3. As shown in Figure 2b, the GFRP-based samples exhibited an EMI shielding effectiveness of approximately 6 dB in maximum, which is lower than that of the CFRP composites, ascribed to the lower electrical conductivity of glass-fiber-based composites, which are regarded as insulators.

### 3.2. Effects of Microfiber Type

The EMI shielding capability of the CFRP and GFRP samples at the representative frequencies is shown in Figure 3, where four representative frequencies (0.7, 2.4, 3.5, and 8.5 GHz) were used.

The frequency point of 0.7 GHz is used in the LTE network and 2.4 GHz is used in microwave ovens, wireless LAN, and Bluetooth [6]. In addition, 3.5 GHz and 8.5 GHz are required in 5G networks and radar surveillance systems, respectively [6]. As shown in Figure 3, the CFRP-based samples demonstrated a much higher EMI shielding effectiveness than the GFRP-based samples, regardless of the quantity and type of embedded CNMs. This is due to the electrical conductivity of the carbon fibers. Therefore, CFRP-based composites are suitable for applications that require protection from unwanted EM waves.

### 3.3. Effects of CNM Contents

The effects of the CNM content on the EMI shielding capabilities of the CFRP- and GFRP-based samples are shown in Figure 4 and Figure 5.

Regardless of the microfiber type, incorporating CNMs improved the EMI shielding capability. Denser electrically conductive networks consisting of CNMs are developed with increasing amounts of CNMs. The reflection and absorption losses increase accordingly [12]. The effects of frequency on the changes in the EMI shielding effectiveness differed based on the microfibers embedded in the FRP composites. No notable changes in the shielding effectiveness were observed for the CFRP-based samples. However, the shielding effectiveness was enhanced with increasing frequency in the GFRP-based samples, where a slight increase in frequency resulted in a considerable increase in the shielding effectiveness of the GFRP samples compared to that of the CFRP samples.

The synergistic effects of the two types of CNMs are shown in Figure 4 and Figure 5, respectively. As shown in these figures, the CNT and graphene samples demonstrated higher EMI shielding effectiveness than the other samples. The effects of the CNM type on improving the shielding effectiveness are discussed in Section 3.4. Based on these results, it can be concluded that incorporating CNMs into GFRP-based composites only improved the shielding effectiveness by 1–4 dB. However, an increase in the shielding effectiveness of 15–20 dB was achieved by adding CNMs to the CFRP-based composites. Thus, CFRP composites fabricated with CNMs are preferable as EMI shielding composites.

### 3.4. Effects of CNM Type

The effects of the CNM type on the EMI shielding effectiveness of the CFRP and GFRP samples are shown in Figure 6.

As seen in the previous sections, the GFRP samples exhibit a frequency effect. In addition, synergistic effects were observed (Figure 6). The combination of CNMs can improve the EMI shielding effectiveness compared to the use of only CNTs. The shielding effectiveness of the CFRP-based samples with only CNTs increased by 7–12 dB compared to that of the pure CFRP composites, whereas increases of 12–20, 13–20, and 15–21 dB were achieved when CNF, graphene, and GNP were combined with CNTs (see Figure 6). In particular, the shape of the CNM affected the increase in the shielding effectiveness, where plate-type CNMs (i.e., graphene and GNP) enabled a more significant improvement than fiber-type CNMs such as CNFs or CNTs. High two-dimensional electrical conductivity was achieved using plate-type conductive fillers (graphene and GNP). The fillers can be easily aligned in the 2D plane during the fabrication process to develop well-formed conductive networks. The advantages of plate-type conductive fillers for improving the EMI shielding effectiveness were addressed by Wang et al. [14], in close agreement with the results of this study.

The EMI shielding effectiveness of the CNM-embedded FRP composites was compared to that of polymer-based composites with CNMs. Maiti and Khatua [23] examined the electromagnetic wave-shielding characteristics of polycarbonate/GNP/CNT composites incorporating 3% GNP/CNT and demonstrated 15 dB of EMI shielding effectiveness at 8 GHz. Liang et al. [24] obtained 21 dB of EMI shielding effectiveness at 8 GHz using epoxy composites incorporating 15 wt.% of graphene [25]. Wang et al., fabricated polyimide foam composites including precipitates of CNT/GO and demonstrated an EMI shielding effectiveness of 28 dB at 8.25 GHz [14]. Al-Saleh et al., studied the influence of the mixing ratios of GNP and CNT on the EMI shielding effectiveness of polypropylene (PP)/polyethylene (PE)-based polymer composites, and obtained an EMI shielding effectiveness of 22 dB at 8 GHz using PP/PE composites with 5 vol.%-GNP/CNT and a GNP to CNT ratio of 1:4 [26]. Considering the EMI shielding effectiveness achieved in these four prior studies, the CFRP composites with 3.0% CNT/GNP in the present study present competitive EMI shielding performance, with an EMI shielding effectiveness of 37.5 dB at 8 GHz. Qian et al. [27] fabricated a Ti_3_C_2_Tx MXene film intercalated with CNF/CNF microspheres and obtained an EMI shielding effectiveness of 45 dB at 8 GHz. The EMI shielding effectiveness of the Ti_3_C_2_Tx/CNF/CNT film was superior to that of the CNM-added CFRP composites in this study. This is attributed to the difference in the base materials of the Ti_3_C_2_Tx/CNF/CNT composites and the CNM-added CFRP composites. The metallic base materials of the Ti_3_C_2_Tx/CNF/CNT composites show superior EMI shielding effectiveness compared with the polymeric base materials of the CNM-added CFRP composites. However, it can be said that various kinds of addition ratio selections are required to analyze and obtain the optimal addition ratio. Thus, further studies will be carried out to obtain the optimal addition ratio, which can lead to the improved EMI shielding performances by using the synergistic effects of various combinations of CNMs observed in the present study.

## 4. Conclusions

In the present study, a comprehensive evaluation of the EMI shielding performance of various types of CNM-embedded CFRP and/or GFRP composites is reported. First, nine combinations of CNMs were added to CFRP and GFRP to fabricate the composites. Subsequently, the synergistic effects of different combinations of CNMs on the EMI shielding performance of the composites were investigated. The key findings can be summarized as follows:(1)The CFRP-based composites achieved much higher EMI shielding performance than the GFRP-based composites. This can be deduced from the high electrical conductivity of micro-typed carbon fiber which composed CFRP.(2)The plate-type CNMs (i.e., GNP and graphene) showed improved synergistic effects compared to the fiber-typed CNMs (CNF).(3)The 3% CNT-GNP CFRP composites, containing plate-typed CNM, exhibited the best EMI shielding effectiveness (38.6 dB at 0.7 GHz).(4)Based on the aforementioned research, the optimal EMI shielding composites based on the optimized combination of CNMs and CFRP/GFRP are expected to be used in trial tests.

The present study focused on the EMI shielding characteristics of the CNMs-embedded FRP composites, and the future research will focus on mechanical properties and durability of the composites.

## Figures and Tables

**Figure 1 polymers-14-05224-f001:**
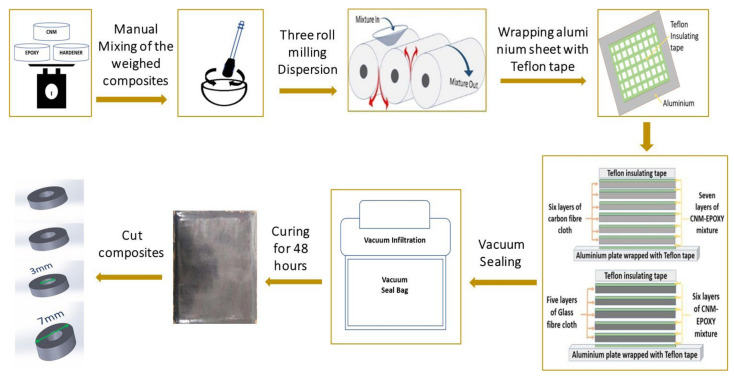
Fabrication of CNM-incorporated CFRP or GFRP composites.

**Figure 2 polymers-14-05224-f002:**
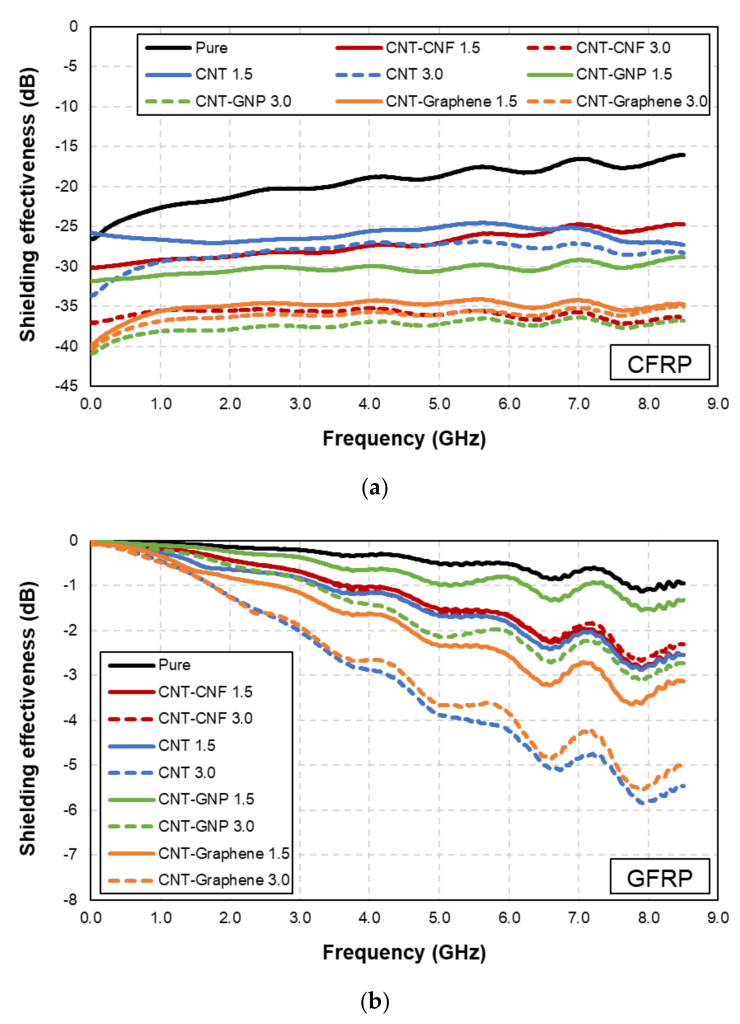
EMI shielding effectiveness of samples fabricated with (**a**) CFRP and (**b**) GFRP matrix.

**Figure 3 polymers-14-05224-f003:**
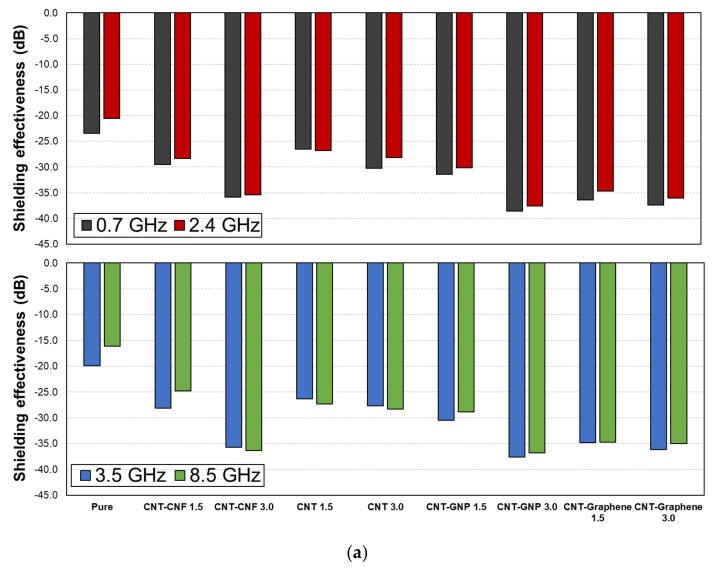
EMI shielding effectiveness of (**a**) CFRP and (**b**) GFRP samples at representative frequencies (0.7, 2.4, 3.5, and 8.5 GHz).

**Figure 4 polymers-14-05224-f004:**
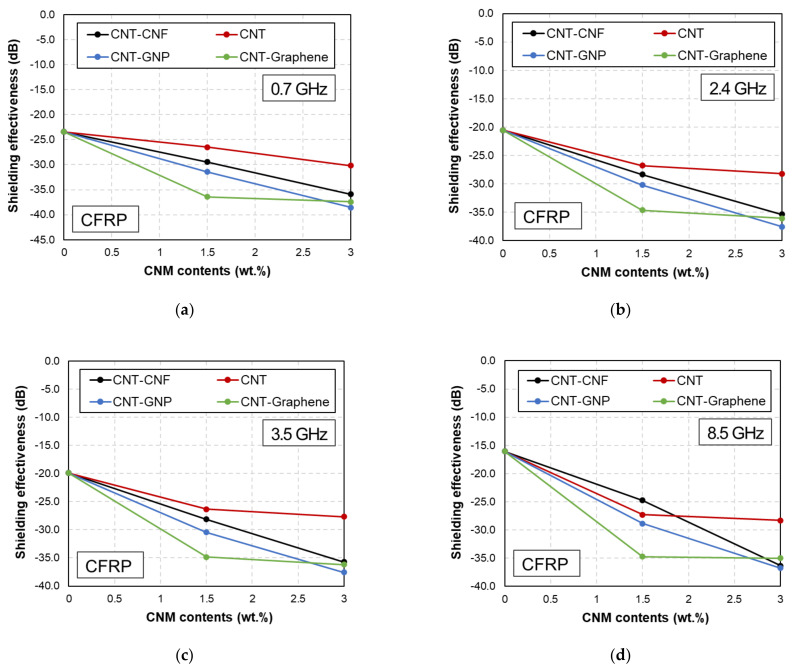
EMI shielding effectiveness of CFRP-based samples with different amounts of CNMs at (**a**) 0.7, (**b**) 2.4, (**c**) 3.5, and (**d**) 8.5 GHz.

**Figure 5 polymers-14-05224-f005:**
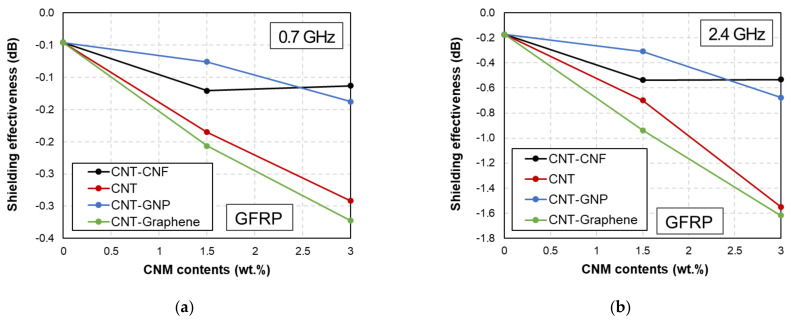
EMI shielding effectiveness of GFRP-based samples with different amounts of CNMs at (**a**) 0.7, (**b**) 2.4, (**c**) 3.5, and (**d**) 8.5 GHz.

**Figure 6 polymers-14-05224-f006:**
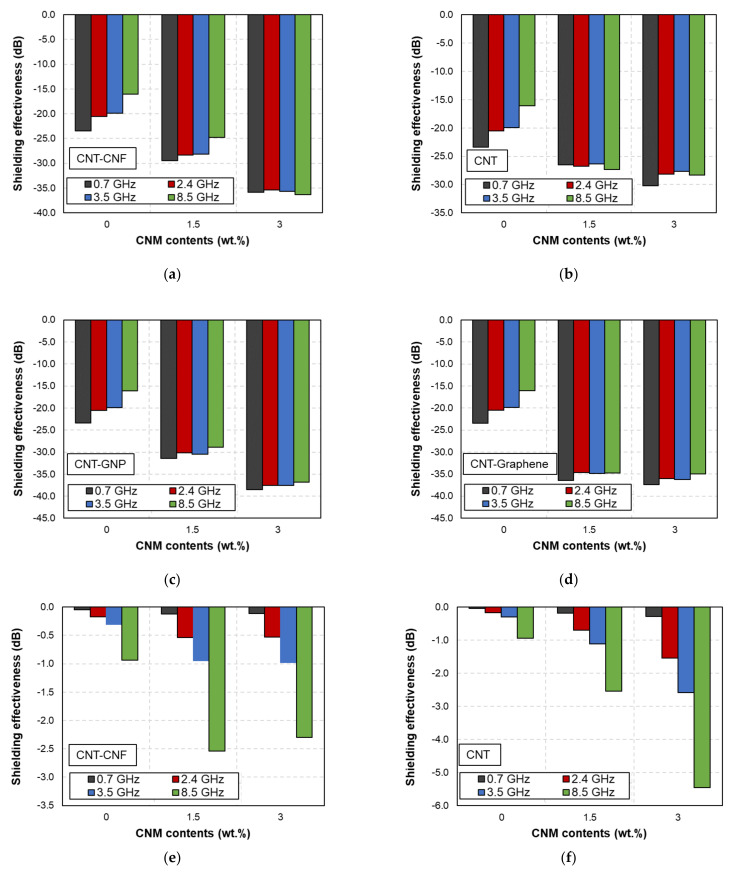
Effects of different CNM types on EMI shielding effectiveness of (**a**–**d**) CFRP- and (**e**–**h**) GFRP-based samples.

**Table 1 polymers-14-05224-t001:** Dimensions and physical properties of carbon nanomaterials.

	CNT	CNF	Graphene	GNP
Diameter or Thickness	8 nm (outer diameter) 2–5 nm (inner diameter)	0.15–0.2 μm	0.55–1.2 nm (thickness)	-
Layers			1–5	<30
Purity	>98%	99.9%	99%	>90%
Specific Surface Area (m^2^/g)	>350	300	>500	
Length or width (μm)	10–30	10–30	0.5–3	2–16
Electrical conductivity (S/cm)	>100	-	184.8	6.67

**Table 2 polymers-14-05224-t002:** Physical properties of carbon and glass fiber fabrics.

Type of Fabric	Carbon Fiber Plain Fabric	Glass Fiber Plain Fabric
Grade (g)	200, 1st level	200, 1st level
Thickness (mm)	<0.11	<0.12
Elongation at break (%)	3	3

**Table 3 polymers-14-05224-t003:** Mix proportions of CFRP- and GFRP-based samples.

Samples	CNMs (g)	Epoxy (g)	Curing Agent (g)	Fiber Vol. % *
Pure CFRP	0	150	50	29.3
Pure GFRP	0			50.8
CNT 1.5% in CFRP	3.05			32.4
CNT 1.5% in GFRP	3.05			22.8
CNT 3.0% in CFRP	6.09			23.1
CNT 3.0% in GFRP	6.09			28.1
CNT-Graphene 1.5% in CFRP	1.525			26.7
CNT-Graphene 1.5% in GFRP	1.525			47.1
CNT-Graphene 3.0% in CFRP	3.045			23.1
CNT-Graphene 3.0% in GFRP	3.045			41.3
CNT-CNF 1.5% in CFRP	1.525			30.0
CNT-CNF 1.5% in GFRP	1.525			33.0
CNT-CNF 3.0% in CFRP	3.045			24.0
CNT-CNF 3.0% in GFRP	3.045			41.3
CNT-GNP 1.5% in CFRP	1.525			29.3
CNT-GNP 1.5% in GFRP	1.525			52.8
CNT-GNP 3.0% in CFRP	3.045			30.0
CNT-GNP 3.0% in GFRP	3.045			45.5

***** Glass fiber or carbon fiber volumetric content ratios were estimated.

## Data Availability

Not applicable.

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
