# Peer review of "A Comprehensive Study on EMI Shielding Performance of Carbon Nanomaterials-Embedded CFRP or GFRP Composites"

_polymers, 2022, doi:10.3390/polym14235224_

Round 1

Reviewer 1 Report

The manuscript having ID: polymers-2040535 which I reviewed evaluates the EMI shielding performance of carbon nanomaterials (CNM) embedded in CFRPs and GFRPs. Although, EMI shielding tests were carried but there is no work on the tensile performance.

The contribution of the manuscript is worthy to be published if tensile tests are added as the samples cutting has already been discussed in the submission. Following are my comments:

1.               What is the source of Table 1 and 2? Please add reference.

2.               In line 131, the samples size were cut as per ASTM standard ASTM D 3039 but there is no discussion on the tensile tests for which these samples were cut.

3.               Include the enhancement or degradation of addition of these nanomaterials on the tensile response of all samples studied. This is article would be incomplete without this study.

4.               Also discuss in biblio. in comparison to other manufacturing techniques that why three-milling is simple? A different procedure of coating the fabric with nanomaterials can also be used. Add a text of pros. and cons. of three-milling fabrication method and pre-coating method as in the following article:

https://doi.org/10.1016/j.coco.2022.101382

5.               Enhance the introduction focusing on the EMI shielding. Cite the following article and the biblio there-in for EMI shielding study of GFRPs and CFRPs.

                                  https://doi.org/10.1016/j.compositesa.2020.105901

Author Response

The manuscript having ID: polymers-2040535 which I reviewed evaluates the EMI shielding performance of carbon nanomaterials (CNM) embedded in CFRPs and GFRPs. Although, EMI shielding tests were carried but there is no work on the tensile performance.

The contribution of the manuscript is worthy to be published if tensile tests are added as the samples cutting has already been discussed in the submission. Following are my comments:

Response: We are sincerely grateful to the reviewer for the elaborated comments. We have responded to each review comment in detail below.

  1. What is the source of Table 1 and 2? Please add reference.

Physical properties of CNMs and micro fiber fabrics were described according to the manufacturers’ specifications. To reflect the reviewer’s comment, the relevant sentences have been newly added in the manuscript as follows.

“The dimensions and physical properties of the CNMs are listed in Table 1. Dimensions, purity, specific surface area, length or width, and electrical conductivity of CNMs shown in Table 1 refers to specifications provided by manufacturers. Carbon and glass-fiber-woven plain fabrics supplied by Miaohan Construction & Technology Co., Ltd. and Suihua Glass Fiber Co., Ltd., respectively, were used to fabricate the CNM-embedded CFRP and GFRP composites. The physical properties of the plain carbon/glass-fiber fabrics are listed in Table 2. The grade, thickness, and elongation at break shown in Table 2 refers to specifications provided by manufacturers.”

  (Lines 87-84, Page 2)

  1. In line 131, the samples size were cut as per ASTM standard ASTM D 3039 but there is no discussion on the tensile tests for which these samples were cut.

The authors also agreed with the reviewer’s comment that mechanical characteristics of the fabricated CNMs-embedded FRP composites are important. However, examinations on the mechanical characteristics of the composites are beyond the scope of the present study and will be subjects of the future studies. To reflect the reviewer’s comment, the relevant sentences have been newly added in the manuscript as follows.

The present study focused on the EMI shielding characteristics of the CNMs-embedded FRP composites, and the future research will focus on mechanical properties and durability of the composites.”

(Lines 303-305, Page 12)

  1. Include the enhancement or degradation of addition of these nanomaterials on the tensile response of all samples studied. This is article would be incomplete without this study.

To reflect the reviewer’s comment, future study plan has been newly added in the revised manuscript. Please refer to the Authors’ Response to comment 2 of Reviewer #1.

  1. Also discuss in biblio. in comparison to other manufacturing techniques that why three-milling is simple? A different procedure of coating the fabric with nanomaterials can also be used. Add a text of pros. and cons. of three-milling fabrication method and pre-coating method as in the following article:

To reflect the reviewer’s comments, the relevant sentences have been newly added in the manuscript.

The three-roll mixer has been used to enhance homogeneous dispersion of nanomaterials in polymer matrix materials, and it has proved increase in reliability of the performance and characteristics of the composites [15]. In addition, this method requires a minimal amount of solvent and improves economic efficiency [19-21]. However, the three-roll milling process may entail disadvantage such as damages in nano-structures of the CNMs due to repeatedly applied mechanical forces.

  1. Enhance the introduction focusing on the EMI shielding. Cite the following article and the biblio there-in for EMI shielding study of GFRPs and CFRPs.

To reflect the reviewer’s comments, the relevant sentences have been newly added in the manuscript.

Din et al. [15[1]] fabricated GNP paper-inserted GFRP composites, and the composite including GNP paper of 240 μm exhibited EMI shielding effectiveness of 59 dB at 8.5 GHz.”

(Lines 60-62, Page 2)

Reviewer 2 Report

This is a good paper based on a nice experimental design. Apart from a general language and style check (the number of sentences starting with "the" is rather high), only minor corrections are required from my perspective:

- In view of the numerous abbreviations (CNT, CNM, CFRP, ...) which are easy to confuse I strongly recommend adding a table of all abbreviations. 

- Table 3 is confusing, too. Please make the fact that in each line two composites are presented clearer.

- The term "three-roll milling machine" is not really correct - it is a three-roll mixer. Milling would imply breakage of particles.

- Some of the writing in Figure 1 is too small to read - please improve.

- The abbreviation "PNA" misses explanation (Acronym finder offers 53 versions).

- Figure 3 is very bulky and could be reduced by plotting all (or maybe only two) frequencies together in one diagram. Similar reductions seem possible for Figures 5 and 6.

- Line 243: "compared to", not "compared with" (I'm well aware that this mistake is so frequent today that it tends to become standard, but it's still wrong).

Author Response

This is a good paper based on a nice experimental design. Apart from a general language and style check (the number of sentences starting with "the" is rather high), only minor corrections are required from my perspective

Response: We are sincerely grateful to the reviewer for the elaborated comments. We have responded to each review comment in detail below.

  1. In view of the numerous abbreviations (CNT, CNM, CFRP, ...) which are easy to confuse I strongly recommend adding a table of all abbreviations. 

All abbreviations used in the present study such as CNT, CNM, CFRP, and GFRP are commonly used in this field. In addition, in the present study, the full name of each abbreviation is already mentioned in the original manuscript.

  1. Table 3 is confusing, too. Please make the fact that in each line two composites are presented clearer.

To reflect the reviewer’s comment, the Table 3 has been revised as follow.

Table 3. Mix proportions of CFRP- and GFRP-based samples.

Samples

CNMs (g)

Epoxy (g)

Curing

agent (g)

Fiber Vol. %*

Pure CFRP

Pure GFRP

0

0

150

50

29.3

50.8

CNT 1.5% in CFRP

CNT 1.5% in GFRP

3.05

3.05

32.4

22.8

CNT 3.0% in CFRP

CNT 3.0% in GFRP

6.09

6.09

23.1

28.1

CNT-Graphene 1.5 % in CFRP

CNT-Graphene 1.5 % in GFRP

1.525

1.525

26.7

47.1

CNT-Graphene 3.0% in CFRP

CNT-Graphene 3.0% in GFRP

3.045

3.045

23.1

41.3

CNT-CNF 1.5% in CFRP

CNT-CNF 1.5% in GFRP

1.525

1.525

30.0

33.0

CNT-CNF 3.0% in CFRP

CNT-CNF 3.0% in GFRP

3.045

3.045

24.0

41.3

CNT-GNP 1.5% in CFRP

CNT-GNP 1.5% in GFRP

1.525

1.525

29.3

52.8

CNT-GNP 3.0% in CFRP

CNT-GNP 3.0% in GFRP

3.045

3.045

30.0

45.5

*Glass fiber or carbon fiber volumetric content ratios were estimated.

  1. The term "three-roll milling machine" is not really correct - it is a three-roll mixer. Milling would imply breakage of particles

To reflect the reviewer’s comment, the relevant sentences have been revised as follows.

“A three-roll milling machine mixer (ZYE-50, Shenzhen Zhong Yi Technology Co., Ltd., China) was used…”

(Lines 117-118, Page 4)

“The utilization of three-roll milling mixer technique is a simple and fast procedure for dispersing CNMs in a polymer matrix.”

(Lines 118-119, Page 4)

“… min and subsequently placed in a three-roll milling machine mixer.”

(Lines 132-133, Page 4)

“…two gaps in the three-roll milling machine mixer …”

(Lines 137-139, Page 6)

  1. Some of the writing in Figure 1 is too small to read - please improve.

To reflect the reviewer’s comment, the relevant figure has been revised as follow.

(Figure 1 of the revised manuscript)

  1. The abbreviation "PNA" misses explanation (Acronym finder offers 53 versions).

To reflect the reviewer’s comment, the relevant sentences have been newly added in the revised manuscript.

“A programmable network analyzer (PNA) network analyzer…”

(Lines 151, Page 5)

  1. Figure 3 is very bulky and could be reduced by plotting all (or maybe only two) frequencies together in one diagram. Similar reductions seem possible for Figures 5 and 6.

To reflect the reviewer’s comment, the relevant figures have been revised as follow.

(a)

(b)

(Figure 3 of the revised manuscript)

  1. Line 243: "compared to", not "compared with" (I'm well aware that this mistake is so frequent today that it tends to become standard, but it's still wrong).

To reflect the reviewer’s comment, the relevant sentences have been revised as follow.

“… FRP composites was compared with to that of polymer-based…”

(Lines 258-259, Page 11)

Reviewer 3 Report

Carbon nanomaterial-embedded CFRP and GFRP composites were prepared and the electromagnetic interference (EMI) shielding performances were investigated. The following comments should be considered and responded to improve the quality of the paper.

1. The current title does not convey the important information related to this paper. It should focus on the composite preparation and shielding performance.

2. Abstract, the authors should provide more mechanism analysis and quantitative results. If the innovation of this paper involves the new materials, the preparation method of this new material should also be further described. In addition, the effect of FRP type on the EMI shielding should be further clarified.

3. This paper adopted CFRP and GFRP as the electromagnetic interference (EMI) shielding composites. Furthermore, the EMI shielding performances of carbon nanomaterial (CNM)-embedded CFRP or GFRP composites were evaluated. However, the reviewer did not see the summary on the performance and advantages of CFRP and GFRP in the introduction, such as light weight and high strength, better corrosion resistance, fatigue resistance and creep resistance compared to metal materials. The authors can review the latest research work below to make necessary supplements. Fatigue & Fracture of Engineering Materials & Structures 42 (5), 1148-1160. Polymers, 2022, 14(12): 2381. Composite Structures 293, 115719.

4. In this paper, vacuum perfusion method is used to prepare composite materials, and different carbon nanoparticles are added into the composite materials with different contents. The related information should also be further summarized to enrich the introduction.

5. The writing of Tables 1 to 3 should be further standardized, with the same font size and format requirements.

6. In table 3 there are only two ratios (1.5% and 3.0%) of carbon nanoparticles in the composite. Finally, the authors concluded that 3.0% was the best addition ratio. Are the two addition ratios adequate to obtain the best performance? Generally speaking, 3-4 kinds of addition ratio selections are desirable to analyze and obtain the optimal addition ratio.

7. The title in Section 2.3 are inappropriate. It is recommended to replace them with EMI shielding performance tests. In addition to the EMI shielding performance test, why not conduct the basic mechanical properties tests of composites, this is significant for the evaluation of material properties in practical engineering applications.

8. For the analysis of EMI shielding effectiveness (figure 2), the conductivity of carbon fiber itself should be taken into account.

9. What is the essential difference between the EMI shielding efficiency for CFRP and GFRP?

10. In part 3.3 and 3.4, data with only three points should not be drawn into a curve. It is suggested to change relevant curves into a column chart to analyze the change trend of EMI shielding efficiency.

11. The conclusions are suggested to condense into 3-4 key information points.

Author Response

Carbon nanomaterial-embedded CFRP and GFRP composites were prepared and the electromagnetic interference (EMI) shielding performances were investigated. The following comments should be considered and responded to improve the quality of the paper.

Response: We are sincerely grateful to the reviewer for the elaborated comment. We have responded to the above comment in detail below

  1. The current title does not convey the important information related to this paper. It should focus on the composite preparation and shielding performance.

To reflect the reviewer’s comment, the title has been revised as follow.

A comprehensive study on EMI shielding performance of carbon nanomaterials-embedded CFRP or GFRP composites

(Title of the revised manuscript)

  1. Abstract, the authors should provide more mechanism analysis and quantitative results. If the innovation of this paper involves the new materials, the preparation method of this new material should also be further described. In addition, the effect of FRP type on the EMI shielding should be further clarified.

To reflect the reviewer’s comment, the relevant sentences have been newly added and revised in the revised manuscript as follow.

The gap distances were carefully adjusted at each cycle, and the gap distance was described as follows. At 1st cycle, the distance of gaps of mixture inlet and outlet were 60 μm and 40 μm, respectively. Then, they were change to 40 μm and 20 μm at 2nd cycle, 20 μm and 15 μm at 3rd cycle, 15 μm and 10 μm at 4th cycle, 10 μm and 5 μm at 5th cycle, 5 μm and 5 μm at 6th cycle, 10 μm and 5 μm at 7th cycle, 5 μm and 5 μm at 8th cycle , 10 μm and 5 μm at 9th cycle, 5 μm and 5 μm at 10th cycle, 3 μm and 2 μm at 11th cycle, and 2 μm and 1 μm at 12th cycle, respectively.

(Lines 135-141, Page 4-5)

Among the eighteen samples, 3% CNT-graphene in CFRP composites, which included plate-typed CNM, exhibited the best EMI shielding performances, showing 38.6 dB at 8.5 GHz.

(Lines 22-23, Page 1)

  1. This paper adopted CFRP and GFRP as the electromagnetic interference (EMI) shielding composites. Furthermore, the EMI shielding performances of carbon nanomaterial (CNM)-embedded CFRP or GFRP composites were evaluated. However, the reviewer did not see the summary on the performance and advantages of CFRP and GFRP in the introduction, such as light weight and high strength, better corrosion resistance, fatigue resistance and creep resistance compared to metal materials. The authors can review the latest research work below to make necessary supplements. Fatigue & Fracture of Engineering Materials & Structures 42 (5), 1148-1160. Polymers, 2022, 14(12): 2381. Composite Structures 293, 115719.

To reflect the reviewer’s comment, the relevant sentences have been newly added and revised in the revised manuscript as follow.

Especially, the investigations of EMI shielding perforamance of carbon/glass-fiber-reinforced polymeric (CFRP and GFRP) composites are rare. These composites have been highlighted due to their light weight, high strength, excellent corrosion resistange, and high fatigue resistance [16[1],17[2]]; however, few studies have reported that EMI shielding performance of CFRP and/or GFRP composites. In addition, comprehensive studies on the effects of the CNM type on the EMI shielding characteristics of CNM-embedded CFRP and GFRP composites have not been examined to date, to the best of our knowledge

(Lines 66-72, Page 2)

  1. In this paper, vacuum perfusion method is used to prepare composite materials, and different carbon nanoparticles are added into the composite materials with different contents. The related information should also be further summarized to enrich the introduction.

To reflect the reviewer’s comment, the relevant sentences have been newly added and revised in the revised manuscript as follow.

“Nine combinations of CNMs and two types of microfibers (i.e., carbon fibers (CFRP) and glass fibers (GFRP)) are used to fabricate the composites to examine the synergistic effects of various combinations of CNMs on improvements of EMI shielding performances

(Lines 74-75, Page 2)

  1. The writing of Tables 1 to 3 should be further standardized, with the same font size and format requirements.

To reflect the reviewer’s comment, the relevant tables have been revised in the revised manuscript.

  1. In table 3 there are only two ratios (1.5% and 3.0%) of carbon nanoparticles in the composite. Finally, the authors concluded that 3.0% was the best addition ratio. Are the two addition ratios adequate to obtain the best performance? Generally speaking, 3-4 kinds of addition ratio selections are desirable to analyze and obtain the optimal addition ratio.

The authors agree with that various kinds of addition ratio selections are required to analyze and obtain the optimal addition ratio. In this study, the authors are try to investigate the synergistic effects of various combinations of CNMs which have different types on improvements of EMI shielding performances. Hence, the further studies will be carried out to obtain the optimal addition ratio which can lead to the improved EMI shielding performances with small amount of CNMs.

To reflect the reviewer’s comment, the relevant tables have been revised in the revised manuscript.

However, it can be said that various kinds of addition ratio selections are required to analyze and obtain the optimal addition ratio. Thus, further studies will be carried out to obtain the optimal addition ratio which can lead to the improved EMI shielding performances by using the synergistic effects of various combinations of CNMs observed in the present study.

(Lines 279-283, Page 12)

  1. The title in Section 2.3 are inappropriate. It is recommended to replace them with EMI shielding performance tests. In addition to the EMI shielding performance test, why not conduct the basic mechanical properties tests of composites, this is significant for the evaluation of material properties in practical engineering applications.

To reflect the reviewer’s comment, the relevant sub-title has been revised as follow.

EMI shielding performance test

(Sub title of section 2.3)

In addition, please refer to the Author’s Response to the Reviewer #1’s comment 2 for further details.

  1. For the analysis of EMI shielding effectiveness (figure 2), the conductivity of carbon fiber itself should be taken into account.

The EMI shielding effectiveness of the pure GFRP or CFRP sample which does not include any CNMs in CFRP can be observed in Figure 2 (a). Thus, the effect of added CNMs on improvements of EMI shielding performances can be obtained by considering the results of pure CFRP or GFRP samples.

  1. What is the essential difference between the EMI shielding efficiency for CFRP and GFRP?

As seen in Figure 2, the pure CFRP sample showed much higher levels of EMI shielding effectiveness than that found in pure GFRP sample. It can be deduced from the microfibers composed of each composites. The micro-typed carbon fiber composed of CFRP which exhibited much higher electrical conductivity than micro-typed glass fiber which composed GFRP. Thus, the samples with CFRP showed improved EMI shielding performances than GFRP samples.

  1. In part 3.3 and 3.4, data with only three points should not be drawn into a curve. It is suggested to change relevant curves into a column chart to analyze the change trend of EMI shielding efficiency.

To reflect the reviewer’s comment, the relevant figures have been revised as follows.

(a)

(b)

(c)

(d)

(e)

(f)

(g)

(h)

(Figure 6 of the revised manuscript)

  1. The conclusions are suggested to condense into 3-4 key information points.

To reflect the reviewer’s comment, the conclusion has been revised as follows.

The key findings can be summarized as follows.

  • The CFRP-based composites achieved much higher EMI shielding performance than the GFRP-based composites. This can be deduced from the high electrical conductivity of micro-typed carbon fiber which composed CFRP.
  • The plate-type CNMs (i.e., GNP and graphene) showed improved synergistic effects compared to the fiber-typed CNMs (CNF).
  • The 3% CNT-graphene composites, containing plate-typed CNM, exhibited the best EMI shielding effectiveness (38.6 dB at 8.5 GHz).
  • Based on the aforementioned research, the optimal EMI shielding composites based on the optimized combination of CNMs and CFRP/GFRP are expected to be used in trial tests.

(Conclusion of the revised manuscript)

[1] C. Li, G. Xian, H. Li, Effect of postcuring immersed in water under hydraulic pressure on fatigue performance of large-diameter pultruded carbon/glass hybrid rod, Fatigue Fract. Eng. Mater. Struct. 42 (2019) 1148–1160. doi:10.1111/ffe.12978

[2] K. Saad, A. Lengyel, Strengthening Timber Structural Members with CFRP and GFRP: A State-of-the-Art Review, Polymers (Basel). 14 (2022). doi:10.3390/polym14122381.

Round 2

Reviewer 1 Report

All my comments have been addressed.

Reviewer 3 Report

Accepted.